# MIRACUM-Pipe: An Adaptable Pipeline for Next-Generation Sequencing Analysis, Reporting, and Visualization for Clinical Decision Making

**DOI:** 10.3390/cancers15133456

**Published:** 2023-07-01

**Authors:** Patrick Metzger, Maria Elena Hess, Andreas Blaumeiser, Thomas Pauli, Vincent Schipperges, Ralf Mertes, Jan Christoph, Philipp Unberath, Niklas Reimer, Raphael Scheible, Anna L. Illert, Hauke Busch, Geoffroy Andrieux, Melanie Boerries

**Affiliations:** 1Institute of Medical Bioinformatics and Systems Medicine, Medical Center-University of Freiburg, Faculty of Medicine, University of Freiburg, 79110 Freiburg, Germany; maria.elena.hess@uniklinik-freiburg.de (M.E.H.); andreas.blaumeiser@uniklinik-freiburg.de (A.B.); thomas.pauli@uniklinik-freiburg.de (T.P.); vincent.schipperges@uniklinik-freiburg.de (V.S.); ralf.mertes@uniklinik-freiburg.de (R.M.); geoffroy.andrieux@uniklinik-freiburg.de (G.A.); 2Faculty of Biology, University of Freiburg, 79104 Freiburg, Germany; 3German Cancer Consortium (DKTK) and German Cancer Research Center (DKFZ), Partner Site Freiburg, 79110 Freiburg, Germany; lena.illert@tum.de; 4Junior Research Group (Bio-)Medical Data Science, Faculty of Medicine, Martin-Luther-University Halle-Wittenberg, 06122 Halle, Germany; jan.christoph@uk-halle.de; 5Medical Informatics, Friedrich-Alexander University Erlangen-Nuremberg, 91058 Erlangen, Germany; philipp.unberath@fau.de; 6Medical Systems Biology Group, Lübeck Institute Für Experimental Dermatology, University of Lübeck, Ratzeburger Alle 160, 23538 Lübeck, Germany; niklas.reimer@uksh.de (N.R.); hauke.busch@uksh.de (H.B.); 7Institute for AI and Informatics in Medicine, University Hospital Rechts der Isar, Technical University Munich, 81675 Munich, Germany; raphael.scheible@tum.de; 8Institute for Immunodeficiency, Center for Chronic Immunodeficiency, Medical Center-University of Freiburg, Faculty of Medicine, University of Freiburg, 79106 Freiburg, Germany; 9Department of Medicine III, Klinikum Rechts der Isar, Faculty of Medicine, Technical University of Munich, 81675 Munich, Germany; 10Department of Medicine I, Medical Center-University of Freiburg, Faculty of Medicine, University of Freiburg, 79110 Freiburg, Germany; 11TranslaTUM, Center for Translational Cancer Research, Technical University of Munich, 81675 Munich, Germany; 12German Cancer Consortium (DKTK) and German Cancer Research Center (DKFZ), Partner Site Munich, 81675 Munich, Germany; 13Center for Personalized Medicine, Klinikum Rechts der Isar, Faculty of Medicine, Technical University of Munich, 81675 Munich, Germany

**Keywords:** molecular tumor board, next-generation sequencing, pipeline, precision oncology, bioinformatics, computational biology, software, workflow, somatic variant calling

## Abstract

**Simple Summary:**

Next-generation sequencing (NGS) is a cutting-edge technology that enables rapid, high-throughput sequencing of DNA and RNA. Researchers and clinicians can identify genetic mutations, gene fusions, and other alterations that may drive cancer growth. This is particularly important in precision oncology as it is applied in the context of Molecular Tumor Boards (MTBs). The latter are multidisciplinary teams of experts who use NGS and bioinformatics tools to analyze patients’ genetic profiles and develop personalized treatment recommendations for cancer patients. Thus, a crucial process for MTB decision-making is the analysis, compilation, and presentation of high-dimensional sequencing data, which are used for both preparation of and case presentation to all stakeholders. MIRACUM-Pipe precisely addresses these requirements and offers an easy-to-use, one-prompt standardized solution to analyze NGS data, including quality control, variant calling, copy number estimation, annotation, visualization, and report generation.

**Abstract:**

(1) Background: Next-generation sequencing (NGS) of patients with advanced tumors is becoming an established method in Molecular Tumor Boards. However, somatic variant detection, interpretation, and report generation, require in-depth knowledge of both bioinformatics and oncology. (2) Methods: MIRACUM-Pipe combines many individual tools into a seamless workflow for comprehensive analyses and annotation of NGS data including quality control, alignment, variant calling, copy number variation estimation, evaluation of complex biomarkers, and RNA fusion detection. (3) Results: MIRACUM-Pipe offers an easy-to-use, one-prompt standardized solution to analyze NGS data, including quality control, variant calling, copy number estimation, annotation, visualization, and report generation. (4) Conclusions: MIRACUM-Pipe, a versatile pipeline for NGS, can be customized according to bioinformatics and clinical needs and to support clinical decision-making with visual processing and interactive reporting.

## 1. Introduction

Molecular precision oncology aims to manipulate or influence a defined, direct, or indirect tumor-specific molecular target. The principle and clinical efficacy of highly effective “molecular targeting” has meanwhile been demonstrated for numerous compounds in large phase III trials [1,2,3,4]. However, the use of approved targeted agents currently remains almost exclusively within histologically defined entities and is often dependent on the evidence of a predictive biomarker. In the concept of molecular oncology, a tumor is not defined exclusively by its histological features but primarily by its molecular (genetic) tumor profile. According to molecular oncology, this profile and its associated biomarkers can be similar across entities and require the same molecular treatment. For example, many targeted compounds with promising results are currently in biomarker-stratified, tumor-agnostic phase I/II trials, providing a glimpse of the future molecular approval landscape. Unfortunately, only a few molecularly driven phase I/II trials are currently available (especially in Europe), so molecular drugs outside of regulatory approval are frequently used “off-label” in individual therapeutic trials. These molecularly driven “off-label” therapy trials are within the scope of Molecular Tumor Boards (MTBs). The main aim of MTBs is to provide therapeutic or diagnostic indications, usually based on genomic analysis, for cancer patients. To this end, data on specific and recurrent molecular mechanisms are collected from many individual patient cases to provide scientific and clinical evidence for the efficacy of therapeutic approaches targeting these mechanisms [5,6,7,8,9]. Eventually, this will bring highly evident molecular therapies into routine clinical care. MTBs consist of a multidisciplinary team that combines medical and scientific expertise with translational oncology, molecular biology, and bioinformatics [10,11]. There are several approaches to expanded molecular genetic analysis, each identifying a different spectrum of genetic alterations. In the context of MTBs, either targeted combinations of genes are studied using targeted NGS (tNGS) approaches to identify specific targets for which there are approved drugs or extended genetic diagnostics are performed using whole exome sequencing (WES), which covers all protein-coding genes, accounting for approximately 1% of the total genome. With the increasing complexity of genetic data available for individual patients, molecular alterations need to be interpreted according to predefined standards and ultimately reviewed in a multidisciplinary MTB. The therapeutic interpretation of molecular data as well as the assessment of pathogenic mutations require not only a high level of scientific interdisciplinarity but also a continuous literature search and poses a great challenge to the MTB team. Due to the complexity of the results, the need for software or supporting tools to interpret and present the results is tremendously high [12,13,14].

To support transparent data integration and decision-making across the MTBs we have developed MIRACUM-Pipe, an automated analysis workflow for NGS, that produces reliable and reproducible results across different facilities.

## 2. Materials and Methods

MIRACUM-Pipe is a workflow that combines many individual tools to create a seamless sequence for comprehensive analysis, annotation, and reporting of NGS data. The workflow mainly consists of several parts: (i) quality control, (ii) alignment, (iii) variant calling, (iv) copy number variation estimation, (v) RNA fusion detection, (vi) annotation, and (vii) reporting and visualization. MIRACUM-Pipe currently supports three different run-modes tailored to the following:WES analysis, i.e., a patient-matched tumor–normal pair of sequencing samples (DNA only),tNGS analysis, i.e., tumor-only sequencing on a hybrid capture-based gene panel (DNA and optional RNA),Tumor-only analysis, i.e., a single tumor sample from WES (DNA only).

Depending on the selected mode different tools, workflows, and parameter settings are used, and the differences are highlighted in the following.

### 2.1. Quality Control

Preprocessing and quality control are performed with FastQC [https://www.bioinformatics.babraham.ac.uk/projects/fastqc; accession: 26 May 2023], Trimmomatic [15], SAMtools [16,17,18], and bedtools [19] to obtain various quality metrics:mean base quality,mean coverage over (exonic) target region,library size,insert size length.

### 2.2. Alignment

The preprocessed sequencing samples are aligned using the Burrows–Wheeler aligner (BWA-MEM) [20] to the reference genome hg19 from UCSC. To obtain high confidence alignments we applied the Genome Analysis Toolkit (GATK) [21] indel realignment, and conducted base quality score recalibration, and duplication removal following the GATK Best Practice recommendations [22,23].

### 2.3. Variant Calling

Somatic variants for WES tumor–normal pairs are identified and filtered for false positives with VarScan2 to obtain high-confidence variant calls. VarScan2 separates the identified variants into somatic, germline, and loss of heterozygosity (LOH). The latter means that a variant is heterozygous in the germline but evolved to a homozygous variant in the tumor.

VarScan2 is similarly used for tumor-only WES samples. The only difference is that no somatic, germline, nor LOH can be distinguished due to the lack of an appropriate control sample. Therefore, the mpileup routines from VarScan2 are used followed by the false positive filter algorithm.

In the case of tNGS, Mutect2 from GATK is used to identify variants followed by the implemented filtering routine to filter and remove false positives. Mutect2 offers the possibility to supply known platform specific sequencing artifacts and a known germline resource to further limit false positive variant calls and help identify potential germline variants. This is particularly useful when no control sample has been analyzed to identify variants that are unique to the tumor.

### 2.4. Copy Number Variation Calling

Copy number variations (CNVs) are identified with Control-FREEC [24,25]. A matched control sample together with GC-content is used for normalization. If no control sample is available, only GC-content is used. Additionally, the tool Sequenza [26] is used to bioinformatically infer tumor purity and ploidy, and with the help of the scarHRD R package [27,28] the HRD score of the sample is inferred. Tumor purity and ploidy results are further used as input parameters for Control-FREEC.

### 2.5. RNA Fusion Calling

RNA fusions are called with the tool FusionCatcher [29] in case RNA sequencing samples are available in tNGS mode.

### 2.6. Annotation

Identified variants are annotated with the tools ANNOVAR [30,31] and SnpEff [32], covering basic gene annotations from either RefSeq or Ensembl, curated databases like gnomAD [33], ClinVar [34], or InterVar [35], and functional annotation sources contained in dbNSFP [36,37] including 38 prediction scores, e.g., REVEL [38] and 8 conservation scores. SnpEff is used to infer the canonical transcript of the alteration.

In addition, cancer genes from OncoKB [39] as well as known hotspot variants [40,41] are highlighted.

Furthermore, the following sample-specific complex biomarkers are calculated.

tumor mutational burden (TMB),homologous recombination deficiency (HRD) score,microsatellite instability (MSI),bioinformatic tumor purity, andploidy.

For TMB calculation, all somatic and protein-coding variants are used. HRD score, consisting of the sum of large-scale transitions (LST) [42], number of telomeric allelic imbalances (TAI) [43], and loss of heterozygosity (HRD-LOH) [44], is inferred, as described above with the R package scarHRD [27] and the Sequenza [26] output as well as ploidy and tumor purity values are also inferred, as described above. Microsatellite status is identified with MSIsensor-pro [45] for tumor–normal pairs and MSIsensor2 [46] for tumor-only cases.

### 2.7. Functional Enrichment Analysis

Functional enrichment analysis with Fisher’s exact test is performed based on either all identified variants, or all genes affected by CNVs to obtain insights into altered pathways. As a signaling pathway source, the Molecular Signatures Database (MSigDB) [47] with the Hallmark gene sets [48] is used. 

### 2.8. Mutational Signature Analysis

A mutational signature analysis is conducted with the R package YAPSA [49] based on the COSMIC signature V2 [50,51,52].

### 2.9. Reporting and Visualization

MIRACUM-Pipe reports all protein-coding and protein-altering variants fulfilling all quality metrics, the variant allele frequency (VAF) and population frequency cutoffs. As default cutoffs, a VAF above 5% and a population frequency below 0.1% are set. However, not only simple variants, such as SNVs and InDels, are reported but also all findings regarding to quality metrics, complex biomarkers, CNVs, and RNA fusions, if applicable. This information is presented in the form of an interactive PDF report that includes hyperlinks to curated online data sources, such as Genome Nexus [53], MetaKB [54] and VarSome [55], which provide background information on variants. Alternatively, the results are written in a data format that can be directly imported into cBioPortal [56,57].

## 3. Results

The results, as well as the appropriate selection and presentation of the MIRACUM-Pipe tools, were carefully chosen and adapted through close collaboration with clinicians and members of the MTBs, based on a comprehensive stakeholder analysis [13].

### 3.1. MIRACUM-Pipe

MIRACUM-Pipe incorporates tools for detecting SNVs, InDels, LOH variants, CNVs, and RNA fusions as well as for determining quality and statistics. Various functional prediction and annotation databases are integrated to annotate the identified variants automatically. The workflow is designed as a fully automated one-prompt solution from the raw sequencing files to the interactive PDF report containing quality assessments, the identified and annotated genetic variations as well as a gene set enrichment analysis of the SNVs and CNVs, respectively. MIRACUM-Pipe consists of bash and R scripts to perform NGS data processing, basic annotation, complex functional annotations, and downstream analysis of the results. The pipeline is divided into three main parts, as shown in Figure 1, namely (1) preprocessing, quality control, and alignment; (2) analysis, annotation, and interpretation, including variant calling and copy number calling; and (3) assembly of results into a PDF report and an input format for visualization in cBioPortal. In addition to identifying genetic alterations, another focus is annotating identified variants and other results from various available database sources to facilitate their interpretation.

#### Performance, Usability, and Configuration

The pipeline is intended to operate on a high-performance computing (HPC) cluster with a minimum number of eight cores, 150 GB of RAM, and 500 GB of hard drive space. After alignment, variant calling and annotation and copy number analysis is run in parallel by default (see Figure 1), saving processing time and distributing resources evenly. As a benchmark example, a WES dataset, consisting of a tumor and a matched germline sample with 100 million paired-end reads each, was analyzed within 12 h on a computer cluster with two 18-Core Intel Xeon E5-2697v4 processors (2.3–3.6 GHz) having 1 TB of RAM. We provide a Docker container to make the installation flawless and error-free, while allowing MIRACUM-Pipe to be distributed to other sites. This container includes a shell script that implements the pipeline processes, certain tools, R libraries, and several databases (Appendix A). Since MIRACUM-Pipe requires additional user-specific databases and files (Appendix A), we designed an environment wrapped around the Docker container (MIRACUM-Pipe-docker). The environment projects the structure inside the Docker container onto the host system, providing an easy solution to add additional databases and tools. It also serves as an interface for data input and output. In addition, the wrapper takes care of the correct Docker syntax when starting the pipeline, and the user can run it as a simple command line tool. This simplifies the application and setup of MIRACUM-Pipe. Due to existing license restrictions, some tools cannot be delivered within the Docker container. To address these issues, our software is split into two GitHub repositories; one for the pipeline itself, which is intended to be used as a Docker container: MIRACUM-Pipe, and another one for its application and setup: MIRACUM-Pipe-docker. The implementation scheme is shown in Figure 2.

### 3.2. MIRACUM-Pipe Results

Results are provided both as an interactive PDF report and as a machine-readable csv-based file that can be seamlessly imported into cBioPortal.

#### 3.2.1. Interactive PDF Report

All identified and annotated variants and copy number variations are automatically compiled into a PDF report. All key results are presented on the first page of the report. These include the TMB, HRD score, and BRCAness (Figure 3A). In addition, the mutations, ranked according to the ACMG classification (InterVar/ClinVar [34,35]), are presented in a tabular form. Further information about a gene can be obtained from hyperlinks to the Genome Nexus database [53], and information about an amino acid exchange can be obtained from the Variant Interpretation for Cancer Consortium Meta-Knowledgebase (VICC) [54], or from the VarSome [55] database (Figure 3B). Quality anomalies are also mentioned on the first page of the report to better assess the results. A detailed presentation of the assessed quality metrics is shown in the report in tabular form with corresponding reference ranges (Figure 4A).

In addition, the report includes the total number of SNVs and InDels, which are further categorized as homozygous or heterozygous variants, and LOH. Mutations are labeled as tumor suppressor genes (TSGs), oncogenes (OGs), or cancer hotspots. They are only considered if they occur in a protein-coding sequence with a population frequency (MAF) below 0.1% and a sample-specific VAF above 5%. However, these thresholds can be adjusted within the pipeline. The identified variants are displayed in a Circos plot (Figure 4B). To better understand the biological processes affected by the variants, a functional enrichment is calculated on hallmark gene sets [48], where terms with the highest significance are reported. In addition to the classical gene set enrichment, the variants are checked against five cancer-associated signaling pathways, namely PI3K-AKT-mTOR, RAF-MEK-ERK, DNA Damage Response, Cell Cycle and Tyrosine Kinases, that play a role in known cancer processes. The lists of genes involved in these pathways were obtained from Qiagen (https://geneglobe.qiagen.com/us/knowledge/pathways; accession: 26 May 2023). The COSMIC mutational signatures, e.g., the BRCAness signature (AC3, DNA damage), which according to Alexandrov et al. [51] provide insight into the selection of therapeutic options, such as PARP inhibitors, calculated using the R package YAPSA [49]. Copy number variations are visualized as an ideogram highlighting the altered regions above the chromosomes (Figure 4C) and explicitly reported for TSGs and Ogs. For a better insight into the processes altered by the chromosomal instabilities, a functional enrichment based on the hallmark gene sets is performed. Furthermore, the quality criteria and coverage are reported. The reports’ appendix lists all detectable somatic variants, LOH, and germline variants that meet all the criteria. Finally, all tools and databases used are listed, including version information.

All provided information related to complex biomarkers, genetic alterations, as well as quality abnormalities, provide clinicians with all the facts necessary to make a comprehensive assessment of the results and prepare a treatment recommendation for discussion within the MTB.

#### 3.2.2. Visualization in cBioPortal

MIRACUM-Pipe reports all findings not only as an interactive PDF report but also in a format for seamless import into cBioPortal. cBioPortal is an open platform for exploring, visualizing, and analyzing multidimensional cancer genomics data and thus supports the translation of large data sets into biological insights and clinical applications [56,57]. For this reason, we have extended cBioPortal according to the needs of the MTB stakeholders [13] within the context of the MIRACUM consortium [60,61,62]. The upload is performed as follows: variants are exported in the mutation annotation format, the copy number variations as discrete copy number values and segmented data, and RNA fusions as structural variants. Additionally, all complex biomarkers and the results of the mutational signature analysis are stored as patient-specific clinical attributes. With all this information at hand, clinicians make extensive use of the features provided by cBioPortal to visualize the mutational landscape (Figure 5) as well as specific variants, including related information on therapeutic options from, e.g., OncoKB [39] (Appendix A). The latter was extended after information on the approval status of a given drug based on the European Medicines Agency (EMA). The extended cBioPortal even offers the possibility to search directly for clinical trials involving the patient’s genetic alterations through an extension of ClincalTrials.gov (Appendix A) [60]. Furthermore, the clinician can use cBioPortal’s powerful virtual trial options to identify similar patients who have already been discussed with the option of documenting therapy recommendations based on the genetic findings generated by MIRACUM-Pipe, thus creating evidence for further similar individual therapeutic trials.

## 4. Discussion

WES analysis is a high-throughput technology that allows the simultaneous sequencing of thousands of genes in a single experiment. In the context of an MTB, WES analysis can provide clinicians with a wealth of information about a patient’s cancer, including potential driver mutations, actionable targets, and drug resistance mechanisms. Therefore, standardized analysis pipelines and visualization strategies are needed to assist all stakeholders involved in an MTB in handling the complex data that will ultimately lead to therapy recommendations for personalized oncology. In this study, we present the design and use of MIRACUM-Pipe for the analysis of NGS and tNGS data and the associated visualization capabilities of these data.

### 4.1. Use and Insights of MIRACUM-Pipe in the Context of an MTB

In general, WES provides a comprehensive tumor genome analysis, identifying known and novel mutations in cancer cells. This information can help clinicians understand the molecular drivers of the tumor and select appropriate treatment options. A WES analysis generates a large amount of data and interpreting the results can be challenging. Clinicians may need specialized training to interpret the data and use it effectively to guide treatment decisions. MIRACUM-Pipe has been developed to support this in a mostly automated way. It combines a thorough selection of useful tools, some with appropriate adaptations, and a dedicated annotation process with various databases. However, the greatest advantage of MIRACUM-Pipe is that all codes and analysis processes are presented transparently and can be adapted flexibly if required. Another advantage of MIRACUM-Pipe is the compilation of the results into a PDF report and an input format for the visualization in cBioPortal. The PDF report can document the analysis and be added to the corresponding medical records. Visualization tools, such as cBioPortal, can significantly impact the MTB by providing a clear and concise representation of complex genomic data. cBioPortal can help to identify patterns and relationships within the data that may not be apparent in a text-based format. Visualization tools help the MTB to communicate findings more effectively to other team members and the patients. The interactive PDF report and cBioPortal facilitate discussion and decision-making processes by presenting data in a clear and concise manner. Other existing pipelines, e.g., [63,64,65,66], do not yet provide a comparatively in-depth and clearly structured presentation of genetic variants, quality metrics, complex biomarkers, CNVs, and RNA fusions in single outputs.

### 4.2. Limitations and Future Directions

The overall flexibility and scalability of the current implementation as a single Docker container is not ideal. Not all features of a current HPC cluster can be used, such as automated queuing and resource distribution. Therefore, we will implement the pipeline in Nextflow’s workflow language [67] in the future. Using a Nextflow workflow offers the possibility of greater flexibility, scalability, and reproducibility, and with Nextflow’s software containers, it becomes easier to exchange tools and incorporate new tools. This modular approach also facilitates the implementation of multiple tools for the same task, such as streamlining the integration of new methodologies. Another technical challenge in using WES data is the quality of the sequencing data. WES analysis requires high-quality DNA samples, and the quality of sequencing data can be affected by several technical factors, such as sequencing depth and coverage. This has to be taken into account in the analysis and has to be manually adjusted in MIRACUM-Pipe accordingly. Similarly, the cost of WES and its analysis must be considered and may currently be a barrier to its widespread use in clinical practice. However, even these technological challenges will dissolve or adapt, so the use of analysis pipelines and visualization tools, such as MIRACUM-Pipe and cBioPortal, will continue to be beneficial. 

Although some of the tools used in MIRACUM-Pipe have been developed using machine learning and artificial intelligence (AI) methods, MTB therapy recommendation currently relies on expert knowledge due to the lack of large enough and well-stratified patient cohorts. One approach to compensate for this lack is few-shot learning, which combines individual patient information with additional data from in vitro screening [68]. This situation will change in the future. Current initiatives, such as PM^4^Onco (Personalized Medicine for Oncology, Medical Informatics Initiative: https://www.medizininformatik-initiative.de/en/node/801, accession: 26 May 2023) aim to harmonize MTB reporting across hospitals to generate a multi-center cohort of genomic and clinical information of MTB cases on which AI methods can be applied for therapy recommendations and response or resistance. Standardized reporting tools, such as MIRACUM-Pipe, will be instrumental in this as they can provide standardized, machine-readable output reports.

## 5. Conclusions

Next-generation sequencing, particularly WES, is increasingly being used to identify new therapeutic options. However, no common standards for analysis strategies, depth, or medical implementation have been established so far. To overcome these problems and to enable easy-to-use analysis, data interoperability and reuse, we have developed MIRACUM-Pipe. This pipeline can be easily adapted to integrate or merge future databases, analysis tools, and workflows. Furthermore, its visualization capabilities as an interactive PDF report and its integration with cBioPortal facilitate the understanding of the complex data for clinicians and all stakeholders in the MTB. 

In this way, MIRACUM-Pipe can support physicians in making personalized therapy recommendations in oncology and unify the standardization of personalized oncology efforts in the German healthcare system.

## Figures and Tables

**Figure 1 cancers-15-03456-f001:**
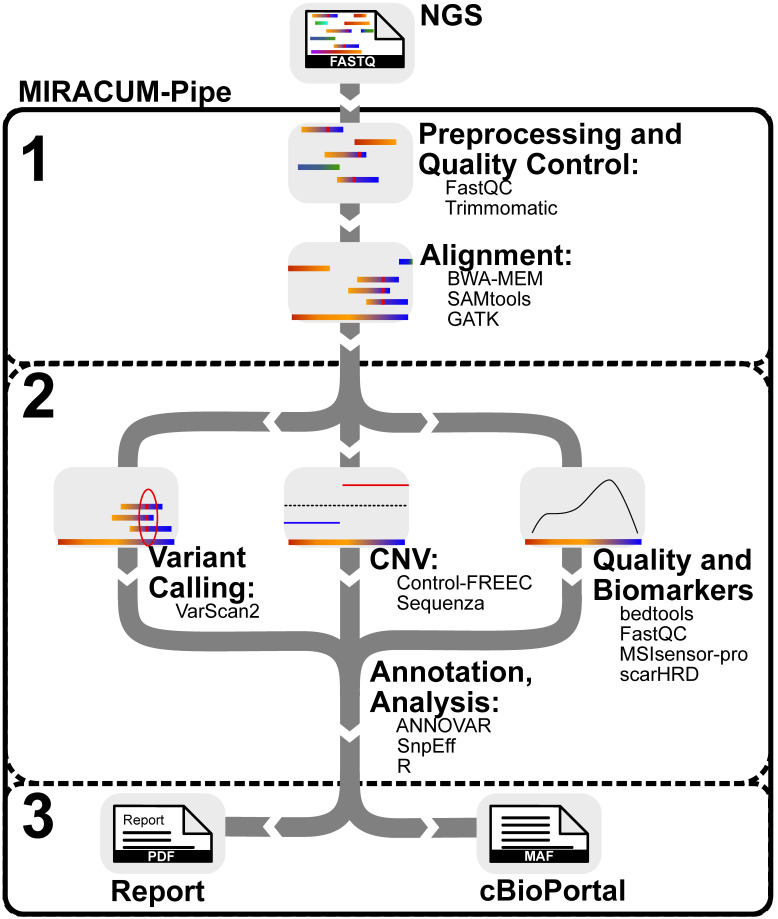
Schematic workflow of MIRACUM-Pipe. MIRACUM-Pipe is divided into three main parts: (1) preprocessing, quality control, and alignment, (2) analysis, annotation, and interpretation, subdivided into variant calling (VC) copy number variations (CNV) calling, and quality assessment and biomarker calculation, and (3) assembly of results.

**Figure 2 cancers-15-03456-f002:**
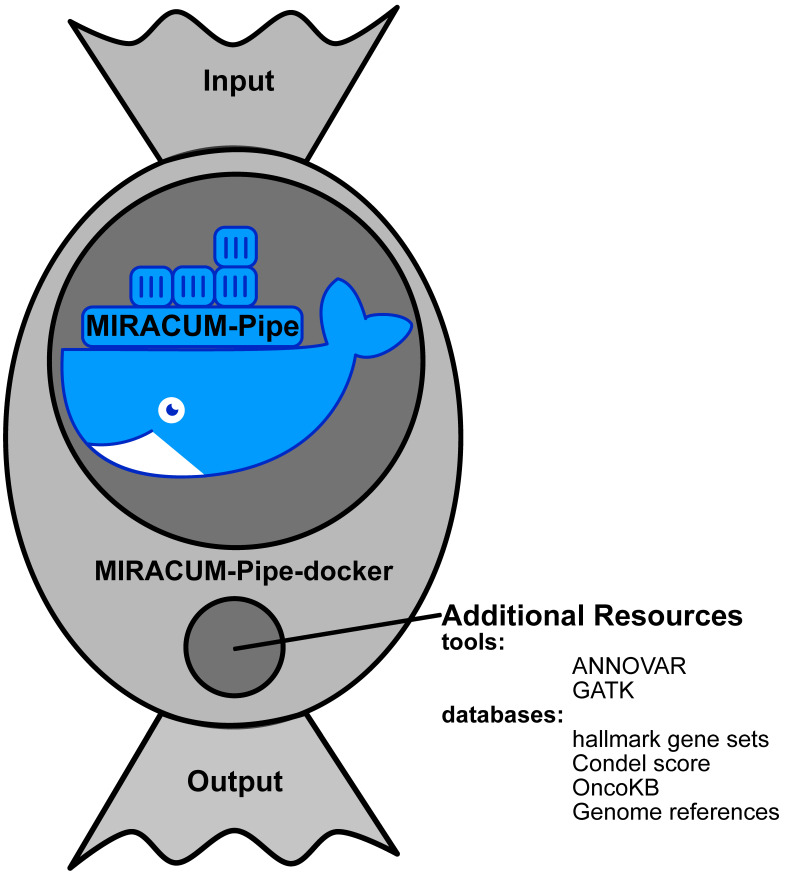
MIRACUM-Pipe implementation scheme with the two available repositories, namely MIRACUM-Pipe and MIRACUM-Pipe-docker. The MIRACUM-Pipe repository builds the Docker container with all necessary source code and most tools, while MIRACUM-Pipe-docker serves as a wrapper to provide additional tools and databases.

**Figure 3 cancers-15-03456-f003:**
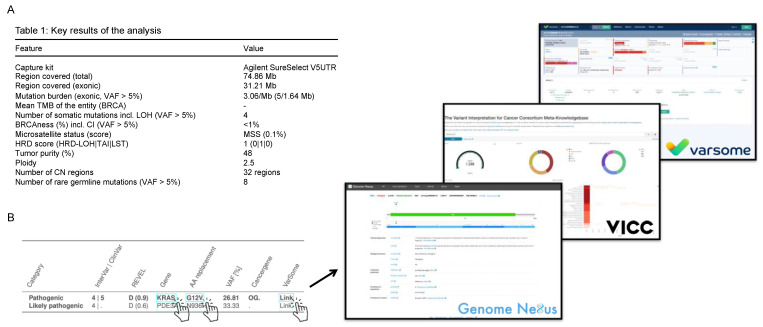
Representations from the first page of the interactive PDF report. (**A**) The most important analysis results are concisely presented. (**B**). Listing of variants and their corresponding link to the database of Genome Nexus [53] (https://www.genomenexus.org/variant/chr12:g.25398284C%3EA; accession: 26 May 2023), the Variant Interpretation for Cancer Consortium Meta-Knowledgebase (VICC) [54] (https://search.cancervariants.org/#KRAS%20G12V; accession: 26 May 2023) and VarSome [55] (https://varsome.com/variant/hg19/chr12%3A25398284%3AC%3AA?; accession: 26 May 2023). The variants are also assigned to the categories of pathogenicity. Further parameters are the REVEL Score [38], the information on the variant allele frequency (VAF), and the assignment to cancer genes (tumor suppressor genes (TSGs) or oncogenes (OGs)).

**Figure 4 cancers-15-03456-f004:**
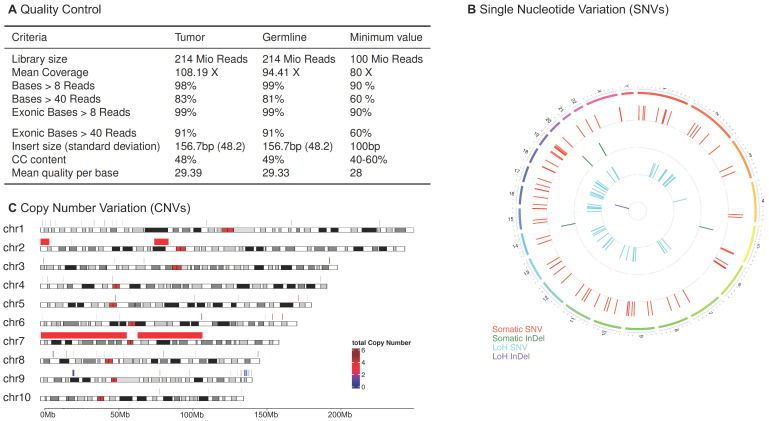
Visual representations of the three key analysis results of MIRACUM-Pipe: (**A**) Quality control (QC) parameters: minimum reference values are provided to assess the sufficiency of the NGS results for the other analyses. The table has been modified to be in English for illustrative purposes, whereas the output language of MIRACUM-Pipe is German, as the report is addressed to clinical experts in Germany. (**B**) Circos plot [58] visualizing the different types of identified variants. Somatic single nucleotide variants (SNVs) are shown in red, somatic insertions and deletions (InDels) in green, loss of heterozygosity (LOH) SNVs in cyan, and (LOH) InDels in purple. Variants are aligned to their actual position on the chromosomes. (**C**) Selection of an ideogram [59] showing copy number variations (CNVs) in the first ten chromosomes of a patient’s tumor genome. The figure briefly assesses which regions are likely to be amplified (red-shaded regions) or lost (blue-shaded regions), indicating increased or decreased activity of genes within that region. A complete table of genes with CNVs is provided in the full report.

**Figure 5 cancers-15-03456-f005:**
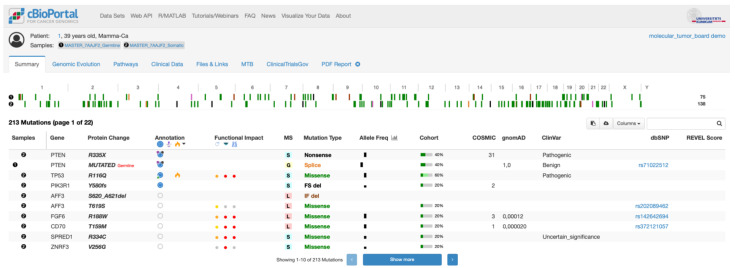
Visualized summary of the mutational landscape in cBioPortal, hosted for the MTB at the Comprehensive Cancer Center Freiburg (CCCF) depicting all SNVs with appropriate structural and additional clinically relevant information. Added features like REVEL score, ClincalTrials.gov, MTB documentation or PDF report can be seen in additional columns and in the tab bar.

## Data Availability

The two repositories are available on GitHub and can be downloaded from https://github.com/AG-Boerries/MIRACUM-Pipe (accession: 26 May 2023) and https://github.com/AG-Boerries/MIRACUM-Pipe-docker (accession: 26 May 2023).

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
