# Peer review of "MIRACUM-Pipe: An Adaptable Pipeline for Next-Generation Sequencing Analysis, Reporting, and Visualization for Clinical Decision Making"

_cancers, 2023, doi:10.3390/cancers15133456_

Round 1

Reviewer 1 Report

In this manuscript, the authors have developed an end-to-end NGS pipeline to support clinical decision-making for cancer care that incorporates many widely used bioinformatic tools. My main comments are below:

1.     Figure 1 should be expanded to include the names of the tools that are used in each step. At its current form, it is too generic and not very informative to the readers.

2.     Figure 2 should be expanded. At its current form, it doesn’t add significant information to the readers to justify its presence. Perhaps a more detailed diagram that explicitly describes the functionalities that each of the two repositories contributes and also the data flow between the additional resources and the input. For example, the figure should indicate what exactly are the input files and also list the additional resources.

3.     Figure 3 should be reworked. At its current state it only offers a glimpse of the tool’s PDF report. The “Genome Nexus” and the “Variant Interpretation for Cancer Consortium Meta-Knowledgebase” screenshots are redundant. The authors should go over in more detail the results of the complete PDF report of MIRACUM-Pipe, how each section contributes to clinical decision-making, and how the results could be interpreted and related to therapy decisions. 

4.     The circus plot for Figure 4A is empty.

5.     Authors should elaborate further on why Figure 4B and C are useful to the expertise consisting of a Molecular Tumor Board. These are more technical/QC metrics rather than actionable results of the pipeline. Perhaps they should be included in a sub-section of the report, but shouldn’t be the highlight of the report.  

6.     The authors need to cover more thoroughly existing automated clinical bioinformatics pipelines (e.g. https://nf-co.re/sarek )and how their solutions differ from existing ones. Perhaps include a comparison table between different pipelines.

7.     My main concern is the usability of the pipeline. While the authors have created a docker container for the pipeline, many of the tools and databases, due to licensing issues, have to be installed locally. This significantly complicates the installation process and cannot ensure that all these tools will be maintained and be accessible in the future. For example, the MIRACUM-Pipe-docker/setup.sh script contains deprecated instructions for installing Annovar. In this case, the link displayed on the instructions doesn’t work anymore (see "please visit http://download.openbioinformatics.org/annovar_download_form.php to get the download link for annovar via an email" on MIRACUM-Pipe-docker/setup.sh).

8.     Another consequence of not having everything on the same docker container is the incompatibility issues with the installation of the R packages. The R script in RScripts/install_packages.R does not specify the versions of each R package that should be installed. Two different users following the exact instructions will end up with a different R environment depending on when they installed it. Since this pipeline is intended to be used by clinical tumor boars, there needs to be extreme attention to the reproducibility of the results that are run in different computational environments.

The article would benefit from a more thorough proofreading. There are quite a few spelling and grammatical errors in both the main text and the figures. E.g. “Additional Ressources” in Figure 2.

Author Response

We would like to thank reviewer 1 for the response to our manuscript "MIRACUM-Pipe: an adaptable pipeline for Next-Generation Sequencing analysis, reporting, and visualization for clinical decision making" (cancers-2350014). Below you will find our point-by-point response and we have marked the corresponding changes in the manuscript in red. We have also edited English language as recommended.

  1. Figure 1 should be expanded to include the names of the tools that are used in each step. At its current form, it is too generic and not very informative to the readers.

Thank you for your valuable comments on the Figure 1. We have completely revised the figure, added the names of the tools, and made it more specific.

  1. Figure 2 should be expanded. At its current form, it doesn’t add significant information to the readers to justify its presence. Perhaps a more detailed diagram that explicitly describes the functionalities that each of the two repositories contributes and also the data flow between the additional resources and the input. For example, the figure should indicate what exactly are the input files and also list the additional resources.

We have completely reworked Figure 2 and improved the illustration of the two repositories. We also added the additional resources required. We took up your comment and tried to visualize the functionality of the wrapper and the data flow. Additionally, we rewrote that part in the main text to explain it better (line 238ff).

  1. Figure 3 should be reworked. At its current state it only offers a glimpse of the tool’s PDF report. The “Genome Nexus” and the “Variant Interpretation for Cancer Consortium Meta-Knowledgebase” screenshots are redundant. The authors should go over in more detail the results of the complete PDF report of MIRACUM-Pipe, how each section contributes to clinical decision-making, and how the results could be interpreted and related to therapy decisions.

Thank you for your suggestions, we understand the importance to show the capabilities of the generated PDF report. However, showing the full report in the manuscript would not be feasible. Yet, to present the full report to the readership, we have deposited a sample PDF report in the repository (see section Data Availability Statement: https://github.com/AG-Boerries/MIRACUM-Pipe-docker). Our intention was to highlight the overview and interactive elements of the PDF report. We hope that we have been able to explain the reasons for this illustration (Figure 3), which we have slightly adapted.

  1. The circus plot for Figure 4A is empty.

We apologize for the inconvenience. Unfortunately, there was a problem converting the Word document to PDF, but we fixed it.

  1. Authors should elaborate further on why Figure 4B and C are useful to the expertise consisting of a Molecular Tumor Board. These are more technical/QC metrics rather than actionable results of the pipeline. Perhaps they should be included in a sub-section of the report, but shouldn’t be the highlight of the report.

We have revised Figure 4 to highlight the most important results of the MIRACUM-Pipe and provide more detailed information in the text (line 272ff). These include the quality metrics of the samples, which play a crucial role in assessing the results and, thus also on the therapy recommendation given in the MTB. We also show the variants found in a circos plot and the copy number changes identified. Both provide the clinician with a quick evaluation of the case.

  1. The authors need to cover more thoroughly existing automated clinical bioinformatics pipelines (e.g. https://nf-co.re/sarek) and how their solutions differ from existing ones. Perhaps include a comparison table between different pipelines.

Thank you for this insightful comment, which we have now tried to address in detail in the manuscript. We have again emphasized the uniqueness of the MIRACUM-Pipe compared to other pipelines by adding a section (line 386ff) after reviewing various existing pipelines and citing the former in the text.

  1. My main concern is the usability of the pipeline. While the authors have created a docker container for the pipeline, many of the tools and databases, due to licensing issues, have to be installed locally. This significantly complicates the installation process and cannot ensure that all these tools will be maintained and be accessible in the future. For example, the MIRACUM-Pipe-docker/setup.sh script contains deprecated instructions for installing Annovar. In this case, the link displayed on the instructions doesn’t work anymore (see "please visit http://download.openbioinformatics.org/annovar_download_form.php to get the download link for annovar via an email" on MIRACUM-Pipe-docker/setup.sh).

We apologize for the inconvenience caused by the incorrect links to ANNOVAR. The corresponding links have been updated and are working properly, and we hope to show the usability of the pipeline as well. We also appreciate your concerns about usability and reproducibility, which we have answered in detail for your last question #8.

  1. Another consequence of not having everything on the same docker container is the incompatibility issues with the installation of the R packages. The R script in RScripts/install_packages.R does not specify the versions of each R package that should be installed. Two different users following the exact instructions will end up with a different R environment depending on when they installed it. Since this pipeline is intended to be used by clinical tumor boars, there needs to be extreme attention to the reproducibility of the results that are run in different computational environments.

We appreciate your concerns about usability and reproducibility. Wherever possible, we have made the process of installing the additional resources as simple as possible. We provide a setup shell script with many necessary steps, such as downloading GATK or the FusionCatcher DB. We also tried to keep the installation process of ANNOVAR simple. The user only needs to have the download link ready; the script does the rest.
Separating the databases, e.g., for ANNOVAR, also allows the user always to use the latest version or the one they prefer. This also simplifies the update process and makes it independent of the Docker container version.
We also see potential limitations due to missing version information, but these are linked to the releases of the container and validated against appropriate test data. This allows us to ensure that the pipeline provides valid and reproducible results. The container releases include the R runtime, R packages, Debian images, and other Debian packages. The container tag uniquely defines these.

Reviewer 2 Report

Dear Authors, I read your manuscript on the tool MIRACUM-Pipe. I would like to congratulate with you for the effort of developing it.

I have just some few comments about the introduction: 

Line 76-77 "MTBs aim to collect data on specific and recurrent molecular mechanisms from numerous individual patient cases to generate scientific as well as clinical evidence on the efficacy of therapeutic approaches targeting these" I would reformulate this sentence to specify that MTBs have the main goal of producing therapeutical or diagnostic indications for patients with cancer that performed genomic analyses, before data collection.

Best regards

Good

Author Response

We appreciate your feedback on our manuscript and also thank you for your valuable comments about the MTB. We have revised the introduction to emphasize more the primary goal of providing therapeutic and/or diagnostic indications for MTBs (line 76ff., we have marked the corresponding changes in red).

Reviewer 3 Report

I am really grateful to review this manuscript. In my opinion, this manuscript can be published once some revision is done successfully. This study introduces the MIRACUM-Pipe, an adaptable pipeline for Next-Generation-Sequencing analysis, reporting, and visualization for clinical decision making, which combines many individual tools to create a seamless workflow for comprehensive analyses and annotation of NGS data covering quality control, alignment, variant calling, copy number variation estimation and RNA-fusion detection. I would like to argue that this is a great achievement. However, this pipeline looks similar with its existing counterpart give that its workflow depends on univariate analysis such as Fisher’s exact test. Machine learning or deep learning can aid in improving its reliability and I would like to ask the authors to address this issue in Discussion. 

Minor editing of English language required. 

Author Response

We would like to thank reviewer 3 for the response to our manuscript "MIRACUM-Pipe: an adaptable pipeline for Next-Generation Sequencing analysis, reporting, and visualization for clinical decision making" (cancers-2350014). We appreciate your valuable comment on the possibility of AI. As suggested, we have expanded the discussion of this aspect (line 407ff, we have marked the corresponding changes in red). The use of AI in MTB decision-making is currently difficult and is based on expert knowledge. This is due to the heterogeneity of cases and a lack of treatment outcome information. In fact, former efforts by the DKFZ to use IBM Watson for MTB decision making have been futile. To improve this situation, German-wide collaborative efforts, such as PM4Onco (Personalized Medicine for Oncology) strive to harmonize MTB reporting with the goal of generating a large data resource of genetic and clinical information. MIRACUM-Pipe will be instrumental in delivering such machine-readable reporting outputs for deep learning and the application of AI.

We have also edited English language as recommended.

Round 2

Reviewer 1 Report

Reviewers have addressed most of my comments. I would recommend filling in the gaps in the Supp. Tables 1-2. For example, in Supp. Table1, Databases Tab, the authors could fill in the empty rows with either the version of the database used or the URL + date/timestamp they accessed the resource. Such edits greatly help with the reproducibility of the tool. 

No concerns

Author Response

Dear Reviewer,

First of all, thank you for taking the time to review our manuscript and for your positive feedback. Also, thank you for the important note about the Supplemental Tables.

We have made the appropriate additions to both Supplemental Tables. You can now also find the versions in the Database tab.

We have uploaded the two Supplemental Tables and also the entire manuscript (the changes are no longer marked in red).

Best regards,

Melanie Börries